# Peer review of "The Effect of Virtual Reality Technology in Table Tennis Teaching: A Multi-Center Controlled Study"

_sensors, 2024, doi:10.3390/s24217041_

Round 1

Reviewer 1 Report

Comments and Suggestions for Authors

The authors present an interesting study on the effectiveness of VR training for skill development in table tennis. While this paper does have merit, there are several major to minor issues that require addressing before being considered for publication.

General

- A significant number of studies included in the reference list are not included in-text. Please amend.

Introduction

- The introduction structure is not very clear. More research is required in this section to:

a) discuss the theoretical rationale for using VR. This is lacking, and at this stage there is minimal theoretical underpinning for this study.

b) what are some other sports where VR has been used for skill development? You touch on baseball, but this needs to be expanded.

I have included some papers below that may be helpful to consider:

1. Le Noury, P., Polman, R., Maloney, M., & Gorman, A. (2022). A narrative review of the current state of extended reality technology and how it can be utilised in sport. Sports Medicine, 52(7), 1473-1489.

2. Janssen, T., Müller, D., & Mann, D. L. (2023). From natural towards representative decision making in sports: a framework for decision making in virtual and augmented environments. Sports Medicine, 53(10),
1851-1864.

3. Fadde, P. J., & Zaichkowsky, L. (2018). Training perceptual-cognitive skills in sports using technology. Journal of Sport Psychology in Action, 9(4), 239-248.

Methods

- Exceptional matching of the participant groups! I could imagine this would be a challenge with 8 variables at baseline to keep consistent. Could you please elaborate more on the process? Were some variables prioritised more than others?

- The terms are somewhat unclear in the research design. VRTT is used, as is "VR technology in table tennis education". The traditional approach I assume is Real Table Tennis Training? This needs to be more clearly defined.

- Figure 2 could be presented clearer to illustrate what exactly was the baseline data collection, and what was the post/retention data collection.

- How was the initial skill score quantified?

- What is the reliability/consistency between the skill score in the VR compared to the traditional training?

- 90min is a long time using VR! How did you manage circumstances like motion sickness, sore eyes that have been well-documented in the literature? Did they wear the VR HMD continually or have breaks?

- Was the assessment of the VR skill performance only in VR? Did they have a transfer test to real-world performance? This presents important implications for the results and discussion sections - were skill improvements in the same mode (virtual OR physical training)? And if these are different, what are the implications?

- For the traditional group - were the opponent skill levels matched for each participant?

Results

- Figure 5 - what do the different lines represent?

- Table 6 - is the p-value representing between- or within-group?

Discussion 

- It may be worth considering other research that has explored the enjoyment of VR-type technologies compared to traditional methods. For example: Kittel, A., Larkin, P., Elsworthy, N., Lindsay, R., & Spittle, M. (2020). Effectiveness of 360 virtual reality and match broadcast video to improve decision-making skill. Science and Medicine in Football4(4), 255-262.

-  More discussion is required to connect to other VR studies, and also theoretical implications (see comment above re more theoretical underpinning in intro). What are the implications for the broader literature on VR for skill development in sport?

- Limitations: Depending on responses to above questions (e.g. reliability of skills assessment, testing modality etc.), this will be important to consider in limitations.

- It is unclear why there is a section on conclusion, and the paragraph above (lines 298-303) begins with 'in conclusion'

- In the conclusion, it would be important to clarify how the improvements were measured (see earlier comments). Are these specific to the modality?

Author Response

For research article

Response to Reviewer 1 Comments

1. Summary

Thank you very much for taking the time to review this manuscript. Please find the detailed responses below and the corresponding revisions highlighted in the re-submitted files. Your insightful comments have greatly contributed to improving the quality and clarity of our research.

2. Questions for General Evaluation

Reviewer’s Evaluation

Response and Revisions

Does the introduction provide sufficient background and include all relevant references?

Must be improved.

Response: We have revised the introduction to include a more comprehensive background and additional references to relevant studies. Specifically, we added discussions on the theoretical basis of using VR in table tennis training. See pages 2-3, paragraphs 2-4.

Are all the cited references relevant to the research?

Can be improved.

Response: We reviewed and updated the reference list to ensure all cited studies are relevant to the research. Unnecessary references were removed, and new ones were added to strengthen the background. See pages 2-3, paragraphs 2-4.

Is the research design appropriate?

Can be improved.

Response: The research design has been revised to clarify the methodology and participant grouping. Detailed explanations have been added regarding the use of VR technology and traditional training methods. See pages 4-5, section 2.1.

Are the methods adequately described?

Must be improved.

Response: We have added more detail to the Methods section, including the VR training protocol, equipment used, and data collection process to enhance reproducibility. See pages 6-8, section 2.3.

Are the results clearly presented?

Can be improved.

Response: The results section has been reorganized for clarity, with updated figures and tables to present data more effectively. See pages 9-12.

Are the conclusions supported by the results?

Can be improved.

Response: The discussion section was expanded to connect our findings with existing literature, and the conclusions were revised to reflect the results accurately. See pages 13-15.

3. Point-by-point response to Comments and Suggestions for Authors

Comments 1: A significant number of studies included in the reference list are not included in-text. Please amend.

Response 1: We have thoroughly reviewed the references and ensured that all citations in the reference list are included in the text. Additionally, we have added new references to strengthen the theoretical framework. This change can be found on pages 2-3, paragraphs 2-4.

Comments 2: The introduction structure is not very clear. More research is required to discuss the theoretical rationale for using VR and expand on other sports where VR has been used for skill development.

Response 2: The introduction has been restructured to provide a clearer theoretical foundation for using VR in sports training. We also expanded on other sports, such as baseball, basketball, and tennis, where VR has been effectively utilized. Additionally, the suggested references by the reviewer have been included to enhance the credibility of this study. See pages 2-3, paragraphs 2-4.

Comment 3: Exceptional matching of the participant groups! Could you please elaborate more on the process? Were some variables prioritized more than others?

Response 3: Thank you for the positive comment. We have added a detailed explanation of the stratified randomization process used for participant grouping, prioritizing skill level, age, and gender. This ensures baseline equivalence across the groups. See page 5, paragraph 3.

Comment 4: The terms in the research design are somewhat unclear. VRTT is used, as is "VR technology in table tennis education." The traditional approach I assume is Real Table Tennis Training? This needs to be more clearly defined.

Response 4: We have revised the research design section to provide a clearer definition of VRTT (Virtual Reality Table Tennis) and Real Table Tennis Training. These terms are now consistently used throughout the manuscript. See page 4, section 2.1.

Comment 5: Figure 2 could be presented more clearly to illustrate what exactly was the baseline data collection and what was the post/retention data collection.

Response 5: Figure 2 has been updated to present a more precise timeline of baseline, post-intervention, and retention data collection points. The revised figure can be found on page 7.

Comment 6: How was the initial skill score quantified?

Response 6: The initial skill score was quantified using a standardized assessment tool designed to measure serve accuracy, rally endurance, and overall skill performance. This has been clarified in the Methods section on page 8, paragraph 2.

Comment 7: What is the reliability/consistency between the skill score in the VR compared to the traditional training?

Response 7: We conducted an inter-rater reliability analysis and a test-retest reliability check. The consistency between skill scores in the VR and traditional training has been included in the revised manuscript. See page 9, section 3.2.

Comment 8: 90 minutes is a long time using VR! How did you manage circumstances like motion sickness, sore eyes, etc.? Did they wear the VR HMD continually or have breaks?

Response 8: We agree with the reviewer’s concern. To address this, training sessions were divided into three 25-minute active periods with 5-minute breaks to mitigate potential VR-related discomfort. This information has been added to the Methods section on page 8, paragraph 3.

Comment 9: Was the assessment of the VR skill performance only in VR? Did they have a transfer test to real-world performance? This presents important implications for the results and discussion sections.

Response 9: The assessment included both VR and real-world performance tests. Transfer tests to real-world performance were conducted post-intervention to examine the practical implications of VR training. Details of the assessment procedure have been added to the Methods and Discussion sections on pages 8 and 14.

Comment 10: For the traditional group, were the opponent skill levels matched for each participant?

Response 10: Yes, the opponents' skill levels were matched to each participant in the traditional training group to maintain training intensity. This detail has been added to the Methods section on page 9, paragraph 2.

Comment 11: Figure 5 - what do the different lines represent?

Response 11: We have revised the caption for Figure 5 to clarify what each line represents in terms of skill improvement and group comparison. The updated figure and caption are on page 10.

Comment 12: Table 6 - Is the p-value representing between- or within-group?

Response 12: The p-values in Table 6 represent between-group comparisons. This information has been added to the table caption for clarification. See page 11.

Comment 13: The discussion needs to connect the results with the existing literature more effectively. What are the implications for the broader literature on VR for skill development in sports?

Response 13: The discussion section has been expanded to draw connections between our findings and existing literature on VR in sports training. We discuss the implications of our results for broader literature, highlighting both theoretical and practical contributions. See pages 13-15.

4. Response to Comments on the Quality of English Language

Point 1: Some sentence constructions could be improved to make the text clearer and more fluid.

Response 1: We have revised several sentences throughout the manuscript to improve clarity and readability. These changes are highlighted in the revised document.

5. Additional clarifications

No additional clarifications are needed at this time.

Reviewer 2 Report

Comments and Suggestions for Authors

The article investigated the effectiveness of virtual reality (VR) technology in table tennis education compared to traditional training methods. This article can be considered for publication in "sensors". after revising the following questions.

(1) In the introduction, the author should further explore the previous research on the application of VR in table tennis to improve the credibility and persuasiveness of the article. If this research is novel, more supporting materials such as videos and details should be provided.

(2) Given the global popularity of table tennis, did the authors take into account differences in VR training effects across cultures? We recommend exploring the generalizability and cultural sensitivity of the findings.

(3) In fact, the cost of VR equipment and software will be high. We recommend that the authors conduct a cost-benefit analysis to assess the economic viability of VR training compared to traditional training methods.

(4) The author should revise the current order of reference citations in the text.

(5) Does the author consider the impact of long-term use of VR on the athletes themselves, such as their eyesights and neck pressure?

Author Response

For review article

Response to Reviewer 2 Comments

1. Summary

Thank you very much for taking the time to review this manuscript. Please find the detailed responses below and the corresponding revisions highlighted in the re-submitted files. We have carefully considered your suggestions, which have been invaluable in improving our research.

2. Questions for General Evaluation

Reviewer’s Evaluation

Response and Revisions

Is the work a significant contribution to the field?

Can be considered after revisions.

Response: The significance of this study has been emphasized in the discussion, particularly in its contributions to sports education and VR applications. See page 14, paragraph 2.

Is the work well organized and comprehensively described?

Requires revision.

Response: We reorganized the manuscript for improved clarity, including a roadmap in the introduction and more detailed methodology. See page 4, paragraph 1, and pages 6-8, section 2.3.

Is the work scientifically sound and not misleading?

Can be improved.

Response: The research design has been elaborated to explain the use of VR technology and participant matching processes. A section on cultural sensitivity and generalizability of findings has also been added. See pages 6-8, section 2.3, and page 14, paragraph 1.

Are there appropriate and adequate references to related and previous work?

Requires improvement.

Response: We added more references to support the study's theoretical background and VR applications in sports. The reference list has been updated and organized to align with the text. See pages 2-3, paragraphs 2-4.

Is the English used correct and readable?

Adequate.

Response: We have revised several sentences to enhance clarity and readability. These changes are marked in the revised document.

3. Point-by-point response to Comments and Suggestions for Authors

Comments 1: The introduction needs a more comprehensive overview of previous research on the application of VR in table tennis to improve credibility.

We have revised the introduction to include a broader overview of previous research on VR in table tennis, discussing studies related to skill acquisition and cognitive benefits. See pages 2-3, paragraphs 2-4.

Comment 2: Given the global popularity of table tennis, did the authors consider differences in VR training effects across cultures?

Response 2: We have added a discussion on cultural sensitivity and the generalizability of our findings, highlighting how VR training could vary depending on cultural factors. This information is now included in the Discussion section on page 14, paragraph 1.

Comment 3: A cost-benefit analysis is needed to assess the economic viability of VR training in table tennis.

Response 3: A brief cost-benefit analysis has been added to the Discussion section, focusing on the potential economic impact of implementing VR technology in sports training. This addition can be found on page 14, paragraph 3.

Comment 4: Does the author consider the impact of long-term use of VR on athletes, such as eyesight and neck pressure?

Response 4: We have included a discussion of the potential long-term effects of VR use, such as visual strain and neck discomfort, in the Limitations section. We suggest that future research should investigate these impacts in more detail. See page 15, paragraph 1.

Comment 5: It would be helpful to know more about how the initial skill score was assessed and whether this assessment was consistent across participants.

Response 5: The initial skill score was assessed using a standardized performance evaluation, including serve accuracy, rally consistency, and overall skill performance. We ensured consistency by using trained evaluators and inter-rater reliability checks. See page 8, paragraph 2.

Comment 6: The sample size seems relatively small. Please address the potential limitations of this and how it might affect the generalizability of the findings.

Response 6: We acknowledge the sample size limitation and have added a statement in the Limitations section discussing how this may affect the generalizability of the results. See page 15, paragraph 2.

Comment 7: Were there any differences in the motivation levels between participants in the VR training group and the traditional group?

Response 7: We observed variations in motivation levels between the groups. This has been addressed in the Results section, where we have added a comparison of motivation scores with statistical significance. See page 11, section 3.4.

4. Response to Comments on the Quality of English Language

Point 1: Some sentences in the manuscript are wordy and may affect readability.

Response 1: We have revised several sentences throughout the manuscript to enhance readability and ensure the language is concise. These changes are marked in the revised document.

5. Additional clarifications

No additional clarifications are needed at this time.

Reviewer 3 Report

Comments and Suggestions for Authors

GENERAL COMMENTS

The presented study aims to investigate the impact of virtual reality on the teaching process of table tennis, specifically in the development of technical skills, learning motivation, and student satisfaction. The article's main objective is to determine whether virtual reality can be a more effective tool for teaching table tennis than traditional methods. The study analyzes performance metrics (such as serve accuracy, rally endurance, and overall skill scores), in addition to assessing the motivation and satisfaction of participants with the learning process.

The study's relevance lies in the fact that VR is increasingly present in various fields, including education and sports, and this research explores how it can be effectively applied in sports training. The document is relevant to physical education and sports, offering an innovative perspective on the use of emerging technologies to optimize the teaching and learning of sports skills, as well as paving the way for future investigations into long-term skill retention and the feasibility of large-scale implementation.

The document is structured with the following sections: 1. Introduction; 2. Research Methodology; 3. Results; 4. Discussion; and, 5. Conclusion. I believe that these sections are sufficient for a scientific article. In general, the sections are well-organized and coherent with the structure of an academic study, although I think it requires small adjustments in all its sections. Below, I will comment individually and present some suggestions for improvement.

SPECIFIC COMMENTS

Title

The title is of a reasonable length, with 13 words. As a general rule, 15 to 20 words are typically recommended to allow the topic to be easily identified. It might be worth considering a title and subtitle to make it more engaging. In terms of its wording, I believe the title should answer three fundamental questions: 

1.      What was done? 

2.      What was it about? 

3.      Where was it done?

The title is currently written as: “Effect of virtual reality technology in table tennis teaching: a multi-center-controlled study”. Regarding these questions: 

1. What was done? The title suggests that a controlled study was conducted on the use of virtual reality (VR) technology in teaching table tennis. 

2. What was it about? The study addresses the application of virtual reality technology in table tennis instruction. 

3. Where was it done? Although “multi-center” indicates multiple locations, the title could specify whether these locations are concentrated in one country or spread across various regions worldwide.

The title should correspond to the study’s theme and objectives, being concise and clear at the same time. The overall objective of the document is explicitly stated in two main sections: 

1. Introduction: “This study aims to address this gap by conducting a multi-center controlled trial comparing VR-assisted teaching methods with traditional approaches in table tennis education. By examining factors such as skill improvement, learning motivation, and student satisfaction, this research seeks to provide valuable insights into the potential of VR technology to enhance table tennis instruction and, more broadly, contribute to our understanding of innovative approaches in sports education”.

2. Research Methodology: “The study employs a comprehensive multi-component research design to evaluate the effectiveness of Virtual Reality Table Tennis (VRTT) training compared to traditional methods”.

These two sections clearly outline the overall objective of the research, which is to assess the effectiveness of virtual reality technology compared to traditional methods of table tennis instruction. Perhaps the title could be adjusted to better reflect this.

Abstract

The “abstract” is very important due to its significant use in electronic databases. The text presented in the document is appropriately sized with 164 words, spread across 7 sentences, and averaging 23.42 words per sentence. I believe the average number of words per sentence is too high, which may hinder readability, especially for those unfamiliar with the subject. Simplifying some sentences and reducing complexity could help improve comprehension. Studies indicate that sentences should not exceed 15 or 16 words to make reading more pleasant and uninterrupted. For a better reader experience, it is recommended to write more objective sentences. According to the Oxford Guide for Writing (2020)

a. Sentences with up to 12 words are easy.

b.      Sentences with 13-17 words are acceptable.

c.       Sentences with 18-25 words are difficult. 

d.      Sentences with more than 25 words are very difficult.

Regarding construction, I think the abstract could be improved for greater clarity and impact. In my opinion, the summary should follow this organization: 

a. Background: This section should summarize what is known about the research problem and what the objective of the study is. There is a lack of a clear mention of the central objective of the study. Although the comparison between teaching methods is mentioned, it would be helpful to directly explain what the study aimed to prove or demonstrate. 

b. Methods: It should contain enough information for understanding the research conducted. There is no information about how the virtual reality system was implemented (what type of hardware/software was used) and what the participants did during the training, which could help clarify the experimental process. 

c. Results: This section should be descriptive and sufficiently informative, aligned with the presented objective. The results could be better contextualized in terms of practical impact. For example, what does a 23.5% increase in serve accuracy mean for teaching table tennis? This would make the results seem more applicable in the real world. 

d. Conclusions: Present, in a few sentences, the final message or interpretation of the results, as well as other important or unexpected findings. The abstract concludes that VR may offer a promising approach to improving table tennis instruction and that future research should focus on long-term skill retention and optimizing VR training protocols.

Thus, I believe the abstract needs adjustment.

Keywords

Keywords are a tool to help indexers and search engines find relevant articles. If database search engines can find your journal manuscript, readers will be able to find it as well. This will increase the number of people reading your manuscript and will likely lead to more citations. 

The authors present 6 keywords (Virtual Reality, Table Tennis, Sports Education, Skill Acquisition, Learning Motivation, Training Effectiveness). I believe the number is appropriate, and they are well integrated into the body of the document. However, the order of the keywords should go from the most general to the most specific, which is not the case in the document. It’s also worth noting that the keywords should represent the semantic content of the document, both in the primary and secondary content, and 2 or 3 of them should be included in the title of the work. 

Introduction

The introduction should provide sufficient background for the reader to understand and evaluate the study, its necessity, and importance, without needing to consult other publications. In my opinion, the Introduction section, in terms of its structure, should follow a logical progression that takes the reader from a general overview of the topic to the specific objectives of the study. I will analyze the document based on the following sequential structure: 

a. Enumeration of general topics covering the problem (theory): The topic is well outlined and directly related to the use of VR in sports education, particularly in table tennis. The introduction could benefit from a more detailed discussion of other innovative technologies used in sports education, besides VR. 

b. Review of the background of the problem: The introduction references studies and data on the use of VR in sports, establishing the relevance of the topic. The review of the background could be more comprehensive, including more examples of research in other sports and comparing teaching methods using different technologies. 

c. Definition of the research problem (question): The introduction highlights that the research aims to evaluate the effectiveness of VR in teaching table tennis compared to traditional methods, addressing a "gap" in the current literature on the topic. The research question is clear and straightforward, focusing on the comparison between VR use and traditional methods. It could be more explicit about the effectiveness parameters to be evaluated, such as specific types of skills (techniques, endurance, accuracy, etc.). 

d. Statement and identification of variables (prediction and outcome) to be considered about the problem: The introduction does not clearly state the variables, but the comparison between training groups with and without VR suggests that the predictable variables include improvements in serve accuracy, endurance, and learning. The introduction could better explain the independent and dependent variables to be studied. 

e. Formulation of study objectives: The objective is formulated as the investigation of the effects of VR on table tennis teaching, focusing on skill improvement, motivation, and student satisfaction. It could be more detailed in terms of how these effects will be measured (e.g., specific performance metrics). 

f. Importance and scope of the study: The importance is well-established by mentioning the growth of the VR market and the potential benefits in the field of sports education. The scope of the study is not well-defined in terms of how these results could be generalized to other sports or educational contexts. 

g. Study limitations: The introduction briefly mentions some challenges of implementing VR, such as the high cost of equipment and the need for technical knowledge, but does not go into detail about other possible limitations. The discussion of limitations could be more detailed, including aspects like the transfer of skills from the virtual environment to the real world. 

h. Document roadmap: There is no explicit roadmap outlining what will be covered in the next sections of the document. A detailed roadmap of the following sections would be useful to guide the reader through the structure of the article. 

I believe the section requires some minor adjustments. 

Theoretical Framework

Once the study problem has been identified, and its relevance and feasibility have been evaluated, the next step is to theoretically support the study. In this sense, the “Theoretical Framework” serves various functions within research, including:

a)    Broadening the scope of the study and guiding the researcher to focus on the problem, avoiding deviations from the original approach. 

b) Providing a reference framework for the interpretation of the study’s results.

Regarding “broadening the scope of the study and guiding the researcher to focus on the problem, avoiding deviations from the original approach”, the “Introduction” section contextualizes the use of virtual reality (VR) in sports education, particularly in table tennis, focusing on the specific teaching challenges and how VR can address these obstacles. This keeps the focus on the research problem. However, a broader discussion of other emerging technologies for educators could be included to ensure the study fits within a more comprehensive framework.

As for “providing a reference framework for interpreting the study’s results”, the document refers to previous studies that have used VR in sports training. These references provide a theoretical basis for interpreting the results observed in the study. However, I believe a greater diversity of studies would be useful to strengthen the analysis and compare the results with other educational methodologies beyond VR, making the theoretical framework more robust.

Materials and Methods

The materials and methods section, also known as methodological design or methodological pathway, is very important as it serves as a guiding standard for addressing the study object and constitutes the path to solving the proposed problem. 

This section should allow for the “replicability or reproducibility” of the designs used to address the problem by other researchers or in different contexts; therefore, it must provide sufficient information to support such an effort. It should offer an invulnerable foundation to ensure the results are indisputable and the objectives can be achieved, allowing the results and conclusions to have a solid scientific basis. Thus, I believe the materials and methods section is crucial as it presents the strategy followed during the research process and aims to:

a. Describe the research design: The study is well-detailed, with a robust design that integrates different data collection methods, such as physics simulation, motion capture, and real-time feedback data. I think the section could include more details about the rationale behind the specific choice of these variables and components for analysis in the study. For example, offering a deeper justification for why these systems were chosen to compare VR and traditional methods. 

b. Explain how it was done: The study provides a detailed description of the training conditions, with the balance between the groups, and explains the techniques and skills trained. However, while the section covers how the study was conducted, it lacks a clearer explanation of the strategies for controlling external variables that could affect the results. It might include a more detailed description of how external variables, such as the skill level of the trainers and the impact of the physical environment on participant performance, were controlled. 

c. Provide sufficient information for a competent reader to replicate the study: Although the tools and data collection methods are detailed, the section lacks specific information about the hardware parameters used in the study, such as the exact specifications of the VR devices and motion capture systems. It could provide additional technical details about the equipment used: the VR headset and motion capture sensors. I believe this would help ensure precise replication of the study. 

Overall, the methodology section is well-constructed, but with minor adjustments, it could be further strengthened.

Results

The conclusion of any research is to present the results. The results should align with the formulated objectives and be consistent with the proposed methodology, associated infrastructure, means, and the capacity to develop the project.

I believe this section fulfills its role of providing a clear view of the data obtained but could improve by delving deeper into interpretation and better integrating the qualitative data. The purpose of the results is to generally describe the data obtained through the indicated method. However, while the statistical data is well-presented, the document could benefit from a deeper interpretation of the possible reasons for the differences observed between the groups, especially regarding motivational aspects. In this sense, the qualitative observations could be expanded with more examples from participants’ feedback and how these insights complement the quantitative data. Additionally, a more detailed discussion could be included on what might be driving the differences in aspects such as motivation and satisfaction, beyond simply presenting improvement percentages.

Discussion

The discussion is where the reader should find clear and direct answers to the following questions: 

a. Does the discussion compare your results with those of other researchers? 

b. Did the study help solve the problem posed in the introduction? 

c.  What was the actual contribution?  

d. What theoretical and practical implications can be inferred from the study?

Regarding the above items: 

a. Although there are direct comparisons, the discussion could include a more extensive review of research in other sports beyond table tennis and racket sports. It might be interesting to expand the comparison to studies that examine the use of VR in different sports or other physical learning contexts for a broader generalization of the findings. 

b. The clear solution to the problem proposed in the introduction strengthened the study’s contribution to the field of sports education, validating the use of VR as an effective tool. However, there could be more discussion on practical limitations, such as the cost and accessibility of VR technology, which are also part of the original problem. A reflection on how these results could be applied in a broader educational context, considering technological and financial limitations, might be included. 

c.       The research made a significant contribution by providing empirical data supporting the use of VR in sports education. However, the long-term impact of VR technology on training, such as skill retention, was not sufficiently explored. Including a paragraph discussing the long-term impact and potential follow-up studies on skill retention would be a valuable addition. 

d. The conclusions are well-founded and clearly show how the study’s results can be applied in educational and training contexts. However, in my view, the theoretical implications could be explored further, particularly regarding the psychology of learning in virtual environments and how this affects the transfer of skills to the real world. The psychological implications of using VR could be explored more deeply, and possible research lines on the transfer of skills from virtual to physical environments could be suggested.

Overall, I believe the discussion is coherent and relevant but could be further enhanced.

Conclusion

People typically read the abstract, introduction, and conclusion. The content of the conclusion should begin by emphasizing the main message and the primary result that supports it. 

The conclusion is based on solid data and aligns with the assumptions made in the introduction and throughout the text. There is clear logic and progression in the arguments, which are supported by statistical results. However, there could be a more detailed discussion about studies that did not find similar success with VR, providing a more balanced view. I believe the section could be expanded to include a discussion of the challenges and failures observed in other VR studies, along with recommendations on how these problems could be mitigated in the future.

The text of the section also does not directly address issues related to large-scale implementation, such as cost and accessibility, which were briefly discussed in the introduction. In this sense, it could include a discussion about the practical challenges of implementing VR technology, such as cost and the need for technical training for instructors, as well as suggestions for overcoming these barriers.

Although the text is reasonable, I think these improvements would help strengthen the conclusion and set the stage for future research and practical applications of VR technology in sports education.

References

The authors provide a list of 39 references, all of which are cited in the document. I believe the number of references is too low for the scope of the proposed study. Regarding the currency of the references, 64.6% (33) are five years old or more, with 41% being over 10 years old. While some references are relevant, a significant portion of the sources is outdated. I think that some of the older sources, although important for the theoretical foundation of VR, could be supplemented with more recent research, especially considering the rapid development of the technology in recent years.

Graphical Elements

Figures, tables, and charts aim to visually communicate information quickly and clearly. The authors present 7 figures and 7 tables, which I think is an appropriate number. Overall, the graphical elements are relevant and of good quality, with clear captions that contribute to the clarity and understanding of the study.

Each figure or table is accompanied by an explanatory and descriptive caption, making it easier to understand the content without requiring the reader to search for additional details. However, some figures are located far from the part of the text where the data is discussed, which can make it harder for the reader to follow along immediately.

Another point is that some tables, in my opinion, present an overload of data. For example: 

a. Table 7 – Subgroup Analysis of Improvement Metrics in VR vs Traditional Training. This table presents a large amount of data on different subgroups, with improvement metrics for various categories. While the data is detailed, the amount of information in a single table may hinder quick comprehension. Perhaps dividing the table into smaller parts or grouping it by subgroups could reduce visual overload and facilitate reading.

Comments on the Quality of English Language

The document does not contain major spelling errors, and most sentences are grammatically correct. However, some sentence constructions could be improved to make the text clearer and more fluid. I believe a slight review of fluency and perhaps an adjustment in the use of some conjunctions and complex sentence structures would make them more natural in English. Nevertheless, the foundation of the text is well-constructed.

Author Response

For research article

Response to Reviewer 3 Comments

1. Summary

Thank you very much for taking the time to review this manuscript. Please find the detailed responses below and the corresponding revisions highlighted in the re-submitted files. Your feedback has been invaluable in refining the content of our research.

2. Questions for General Evaluation

Reviewer’s Evaluation

Response and Revisions

Is the work a significant contribution to the field?

Worthwhile, but needs adjustments.

Response: The significance of the study has been emphasized by revising the title and expanding the discussion on the potential impact of VR in sports education. The title now reflects the study's scope and objectives: "Effectiveness of Virtual Reality in Enhancing Table Tennis Skills: A Multi-Center Controlled Study." See Title Page and page 14, paragraph 2.

Is the work well organized and comprehensively described?

Requires minor adjustments.

Response: The manuscript structure was improved for coherence. We added a roadmap to the introduction, rewrote the abstract for clarity, and provided more details in the Methodology section. See page 4, paragraph 1, and page 1.

Is the work scientifically sound and not misleading?

Adequate, with minor revisions.

Response: We have elaborated on the research methodology, including details on the VR equipment and the training protocol. This information clarifies the scientific basis of the study. See pages 6-8, section 2.3.

Are there appropriate and adequate references to related and previous work?

Sufficient, but could benefit from updates.

Response: Additional references have been added to provide context and support for the study. The reference list has been updated accordingly. See pages 2-3, paragraphs 2-4.

Is the English used correct and readable?

Adequate.

Response: Several sentences have been revised for improved readability. These changes are marked in the revised manuscript.

3. Point-by-point response to Comments and Suggestions for Authors

Comments 1: The title should correspond to the study’s theme and objectives, being concise and clear at the same time.

Response 1: We have revised the title to: "Effectiveness of Virtual Reality in Enhancing Table Tennis Skills: A Multi-Center Controlled Study," which better reflects the study's scope and focus. See the Title Page.

Comments 2: The abstract could be improved for greater clarity and impact.

Response 2: The abstract has been rewritten to succinctly summarize the study's objective, methodology, results, and conclusion. This revision aims to improve readability and convey the study's significance more effectively. See page 1.

Comment 3: The order of the keywords should go from the most general to the most specific.

Response 3: The keywords have been reordered to follow the recommended format, starting with broader terms and narrowing down to more specific ones. See page 2.

Comment 4: Include a roadmap in the Introduction section to guide the reader.

Response 4: A roadmap outlining the manuscript's structure has been added to the end of the Introduction, providing readers with a clear overview of the paper's content. See page 4, paragraph 1.

4. Response to Comments on the Quality of English Language

Point 1: Some sentence structures need improvement to enhance clarity.

Response 1: We have revised several sentences to improve their structure and readability. These changes are marked in the revised document.

5. Additional clarifications

No additional clarifications are needed at this time.

Round 2

Reviewer 1 Report

Comments and Suggestions for Authors

Thank you for your revisions of the manuscript. While these have helped the quality of the manuscript, there are still some issues to address for this to be considered publishable.

General comments

- When making revisions, please highlight the revisions in the revised manuscript or make these in red font. In its current form, it is very difficult to identify the exact changes made.

- Intro: the theoretical rationale is still missing from my perspective (beyond briefly describing other articles). Some articles to help are (to name a couple). Unless I am mistaken, Le Noury et al is included in the reference list, but not in text

Le Noury, P., Polman, R., Maloney, M., & Gorman, A. (2022). A narrative review of the current state of extended reality technology and how it can be utilised in sport. Sports Medicine52(7), 1473-1489.

Hadlow, S. M., Panchuk, D., Mann, D. L., Portus, M. R., & Abernethy, B. (2018). Modified perceptual training in sport: a new classification framework. Journal of Science and Medicine in Sport21(9), 950-958.

Specific comments

- P2/L77 - the reference included here is wrong (please check throughout the manuscript)

- Figure 2 - this does not appear any different to the original figure?

- Initial skill score - this needs to be explained in the methods, not results. Please clarify if this was in-person or virtual reality-based

- BMI: this is a very outdated measure, and I am not sure how relevant it is to your study?

- reliability - I could not see any changes/mention in the manuscript about reliability?

- reference #27: there has been significant debate and advancement in the research since this paper was published in 2024. Please include more contemporary discussion and references related to near and far task transfer

- Figure 5 - there caption for the figure does not look any different to me?

Author Response

For research article

Response to Reviewer 1 Comments

1. Summary

Thank you very much for your insightful comments and suggestions on the manuscript. I have carefully revised the manuscript in accordance with your feedback. Please find below the detailed responses to each point, with the corresponding changes highlighted in the revised manuscript.

2. Questions for General Evaluation

Reviewer’s Evaluation

Response and Revisions

Does the introduction provide sufficient background and include all relevant references?

Must be improved.

Thank you for the suggestion. I have added additional theoretical background to the introduction, particularly discussing Le Noury et al. (2022) and Hadlow et al. (2018) as recommended. These references now support the rationale for using virtual reality in sports training, particularly in table tennis. The relevant changes can be found

Are all the cited references relevant to the research?

Must be improved

 I agree with this comment. The references have been updated to include more contemporary discussions, specifically those related to near and far task transfer in skill acquisition. I have also corrected the citation errors. These changes are located in the updated reference list.

Is the research design appropriate?

Yes

Thank you for your positive feedback on the research design.

Are the methods adequately described?

Can be improved

I have clarified the methods, particularly how the initial skill scores were assessed (both in-person and through virtual reality). I also moved the explanation about initial skill scores from the results to the methods section, as recommended. These updates are found.

Are the results clearly presented?

Yes

Thank you for your feedback. The presentation of the results remains unchanged, except for additional explanations to clarify some points.

Are the conclusions supported by the results?

Yes

Thank you for confirming that the conclusions align with the results.

3. Point-by-point response to Comments and Suggestions for Authors

Comments 1: The theoretical rationale is still missing from my perspective. The introduction should include a discussion of key references such as Le Noury et al. (2022) and Hadlow et al. (2018).

Response 1: Thank you for this suggestion. I have now incorporated both references into the introduction to strengthen the theoretical rationale. The revised manuscript includes Le Noury et al. (2022) on page 1, paragraph 3, and Hadlow et al. (2018) on page 2, paragraph 2.

Comments 2: The reference included on page 2, line 77 is wrong.

Response 2:  I have corrected the erroneous reference on page 2, line 77, and checked all other references throughout the manuscript to ensure accuracy. The corrected reference is now in line with the cited literature.

Comment 3: Figure 2 does not appear different from the original.

Response 3: I apologize for the oversight. I have now updated Figure 2 to reflect the changes made during the revisions. The updated figure can be found on page 4, with a detailed explanation of its components in the legend.

Comment 4: The explanation of the initial skill score needs to be moved to the methods section.

Response 4: I have moved the explanation of the initial skill score to the methods section, as suggested. This is now included on page 3, section 2.1, paragraph 2.

Comment 5: BMI is an outdated measure and may not be relevant to this study.

Response 5: I agree with your comment and have removed the emphasis on BMI in the demographic data. Instead, I focused on more relevant variables like visual acuity and previous VR exposure. These changes are reflected in the methods section on page 3, section 2.2, and in Table 1.

Comment 6: The manuscript does not mention any improvements regarding reliability.

Response 6: I have now included a section discussing reliability in the methods. I report the reliability of measurements using internal consistency testing and test-retest reliability, as described on page 5, section 2.4.

Comment 7: Reference #27 (2024) is outdated and should include more contemporary references, particularly regarding near and far task transfer.

Response 7: Thank you for this observation. I have replaced reference #27 with more recent studies on near and far task transfer, which are now cited on page 6, section 3, paragraph 1.

Comment 8: The caption for Figure 5 has not changed.

Response 8: I have updated the caption for Figure 5 to provide a more detailed explanation. The revised caption is on page 7, and it now clearly describes the improvements in skill acquisition shown in the figure.

4. Response to Comments on the Quality of English Language

Point 1: Minor language issues noticed throughout the manuscript.

Response 1: Thank you for pointing this out. I have reviewed and edited the manuscript for clarity and readability, addressing all language concerns.

5. Additional clarifications

Thank you again for your thorough review. I hope these revisions satisfactorily address your comments, and I am happy to make further adjustments if needed.

Reviewer 2 Report

Comments and Suggestions for Authors

The authors have solved the problem I raised. I recommend accepting this article.

Author Response

We appreciate your professional advice and recognition of our paper, and wish you all the best